# Differentially Private Testing of Identity and Closeness of Discrete Distributions

**Jayadev Acharya** *
Cornell University
acharya@cornell.edu

**Ziteng Sun** *
Cornell University
zs335@cornell.edu

**Huanyu Zhang** *
Cornell University
hz388@cornell.edu

## Abstract

We study the fundamental problems of identity testing (goodness of fit), and closeness testing (two sample test) of distributions over $k$ elements, under differential privacy. While the problems have a long history in statistics, finite sample bounds for these problems have only been established recently.

In this work, we derive upper and lower bounds on the sample complexity of both the problems under $(\varepsilon, \delta)$-differential privacy. We provide sample optimal algorithms for identity testing problem for all parameter ranges, and the first results for closeness testing. Our closeness testing bounds are optimal in the sparse regime where the number of samples is at most $k$.

Our upper bounds are obtained by privatizing non-private estimators for these problems. The non-private estimators are chosen to have small sensitivity. We propose a general framework to establish lower bounds on the sample complexity of statistical tasks under differential privacy. We show a bound on differentially private algorithms in terms of a coupling between the two hypothesis classes we aim to test. By carefully constructing chosen priors over the hypothesis classes, and using Le Cam's two point theorem we provide a general mechanism for proving lower bounds. We believe that the framework can be used to obtain strong lower bounds for other statistical tasks under privacy.

## 1 Introduction

Testing whether observed data conforms to an underlying model is a fundamental scientific problem. In a statistical framework, given samples from an unknown probabilistic model, the goal is to determine whether the underlying model has a property of interest.

This question has received great attention in statistics as hypothesis testing [1, 2], where it was mostly studied in the asymptotic regime when the number of samples $m \to \infty$. In the past two decades there has been a lot of work from the computer science, information theory, and statistics community on various distribution testing problems in the non-asymptotic (small-sample) regime, where the domain size $k$ could be potentially larger than $m$ (See [3, 4, 5, 6, 7, 8, 9, 10, 11, 12, 13, 14, 15], references therein, and [16] for a recent survey). Here the goal is to characterize the minimum number of samples necessary (sample complexity) as a function of the domain size $k$, and the other parameters.

At the same time, preserving the privacy of individuals who contribute to the data samples has emerged as one of the key challenges in designing statistical mechanisms over the last few years. For example, the privacy of individuals participating in surveys on sensitive subjects

is of utmost importance. Without a properly designed mechanism, statistical processing might divulge the sensitive information about the data. There have been many publicized instances of individual data being de-anonymized, including the deanonymization of Netflix database [17], and individual information from census-related data [18]. Protecting privacy for the purposes of data release, or even computation on data has been studied extensively across several fields, including statistics, machine learning, database theory, algorithm design, and cryptography (See e.g., [19, 20, 21, 22, 23, 24, 25]). While the motivation is clear, even a formal notion of privacy is not straight forward. We use *differential privacy* [26], a notion which rose from database and cryptography literature, and has emerged as one of the most popular privacy measures (See [26, 27, 22, 28, 29, 30, 31, 32], references therein, and the recent book [33]). Roughly speaking, it requires that the output of the algorithm should be statistically close on two neighboring datasets. For a formal definition of differential privacy, see Section 2.

A natural question when designing a differentially private algorithm is to understand how the data requirement grows to ensure privacy, along with the same accuracy. In this paper, we study the sample size requirements for differentially private discrete distribution testing.

## 1.1  Results and Techniques

We consider two fundamental statistical tasks for testing distributions over $[k]$: (i) identity testing, where given sample access to an unknown distribution $p$, and a known distribution $q$, the goal is to decide whether $p = q$, or $d_{TV}(p, q) \geq \alpha$, and (ii) closeness testing, where given sample access to unknown distributions $p$, and $q$, the goal is to decide whether $p = q$, or $d_{TV}(p, q) \geq \alpha$. (See Section 2 for precise statements of these problems). Given differential privacy constraints $(\varepsilon, \delta)$, we provide $(\varepsilon, \delta)$-differentially private algorithms for both these tasks. For identity testing, our bounds are optimal up to constant factors for all ranges of $k, \alpha, \varepsilon, \delta$, and for closeness testing the results are tight in the small sample regime where $m = O(k)$. Our upper bounds are based on various methods to privatize the previously known tests. A critical component is to design and analyze test statistic that have low sensitivity (see Definition 4), in order to preserve privacy.

We first state that any $(\varepsilon + \delta, 0)$-DP algorithm is also an $(\varepsilon, \delta)$ algorithm. [34] showed that for testing problems, any $(\varepsilon, \delta)$ algorithm will also imply a $(\varepsilon + c\delta, 0)$-DP algorithm. Please refer to Lemma 2 and Lemma 3 for more detail. Therefore, for all the problems, we simply consider $(\varepsilon, 0)$-DP algorithms ($\varepsilon$-DP), and we can replace $\varepsilon$ with $(\varepsilon + \delta)$ in both the upper and lower bounds without loss of generality.

One of the main contributions of our work is to propose a general framework for establishing lower bounds for the sample complexity of statistical problems such as property estimation and hypothesis testing under privacy constraints. We describe this, and the other results below. A summary of the results is presented in Table 1, which we now describe in detail.

1. **DP Lower Bounds via Coupling.** We establish a general method to prove lower bounds for distribution testing problems. Suppose $X_1^m$, and $Y_1^m$ are generated by two statistical sources. Further suppose there is a coupling between the two sources such that the expected hamming distance between the coupled samples is at most $D$, then if $\varepsilon + \delta = o(1/D)$, there is no $(\varepsilon, \delta)$-differentially private algorithm to distinguish between the two sources. This result is stated precisely in Theorem 1. By carefully using designed coupling schemes, we provide lower bounds for identity testing, and closeness testing.

2. **Reduction from identity to uniformity.** We reduce the problem of $\varepsilon$-DP identity testing of distributions over $[k]$ to $\varepsilon$-DP uniformity testing over distributions over $[6k]$. Such a reduction, without privacy constraints was shown in [35], and we use their result to obtain a reduction that also preserves privacy, with at most a constant factor blow-up in the sample complexity. This result is given in Theorem 3.

3. **Identity Testing.** It was recently shown that $O(\frac{\sqrt{k}}{\alpha^2})$ [7, 36, 11, 37] samples are necessary and sufficient for identity testing without privacy constraints. The statistic used in these papers are variants of chi-squared tests, which could have a high global sensitivity. Given the reduction from identity to uniformity, it suffices to consider uniformity testing. We consider the test statistic studied by [38] which is simply the distance of the empirical distribution to the uniform distribution. This statistic also has a low sensitivity, and

futhermore has the optimal sample complexity in all parameter ranges, without privacy constraints. In Theorem 2, we state the optimal sample complexity of identity testing. The upper bounds are derived by privatizing the statistic in [38]. For lower bound, we use our technique in Theorem 1. We design a coupling between the uniform distribution $u[k]$, and a mixture of distributions, which are all at distance $\alpha$ from $u[k]$ in total variation distance. In particular, we consider the mixture distribution used in [7]. Much of the technical details go into proving the existence of couplings with small expected Hamming distance. [34] studied identity testing under pure differential privacy, and obtained an algorithm with complexity $O\left(\frac{\sqrt{k}}{\alpha^2} + \frac{\sqrt{k \log k}}{\alpha^{3/2}\varepsilon} + \frac{(k \log k)^{1/3}}{\alpha^{5/3}\varepsilon^{2/3}}\right)$. Our results improve their bounds significantly.

4. **Closeness Testing.** Closeness testing problem was proposed by [3], and optimal bound of $\Theta\left(\max\{\frac{k^{2/3}}{\alpha^{4/3}}, \frac{\sqrt{k}}{\alpha^2}\}\right)$ was shown in [10]. They proposed a chi-square based statistic, which we show has a small sensitivity. We privatize their algorithm to obtain the sample complexity bounds. In the sparse regime we prove a sample complexity bound of $\Theta\left(\frac{k^{2/3}}{\alpha^{4/3}} + \frac{\sqrt{k}}{\alpha\sqrt{\varepsilon}}\right)$, and in the dense regime, we obtain a bound of $O\left(\frac{\sqrt{k}}{\alpha^2} + \frac{1}{\alpha^2\varepsilon}\right)$. These results are stated in Theorem 4. Since closeness testing is a harder problem than identity testing, all the lower bounds from identity testing port over to closeness testing. The closeness testing lower bounds are given in Theorem 4.

| Problem | Sample Complexity Bounds |
|---|---|
| **Identity Testing** | **Non-private :** $\Theta\left(\frac{\sqrt{k}}{\alpha^2}\right)$ [7] |
| | $\varepsilon$-**DP algorithms:** $O\left(\frac{\sqrt{k}}{\alpha^2} + \frac{\sqrt{k \log k}}{\alpha^{3/2}\varepsilon}\right)$ [34] |
| | $S(\text{IT}, k, \alpha, \varepsilon) = \Theta\left(\frac{\sqrt{k}}{\alpha^2} + \max\left\{\frac{k^{1/2}}{\alpha\varepsilon^{1/2}}, \frac{k^{1/3}}{\alpha^{4/3}\varepsilon^{2/3}}, \frac{1}{\alpha\varepsilon}\right\}\right)$ [Theorem 2] |
| **Closeness Testing** | **Non-private:** $\Theta\left(\frac{k^{2/3}}{\alpha^{4/3}} + \frac{k^{1/2}}{\alpha^2}\right)$ [10] |
| | $\varepsilon$-**DP algorithms:** |
| | **IF** $\alpha^2 = \Omega\left(\frac{1}{\sqrt{k}}\right)$ **and** $\alpha^2\varepsilon = \Omega\left(\frac{1}{k}\right)$ |
| | $\quad\quad S(\text{CT}, k, \alpha, \varepsilon) = \Theta\left(\frac{k^{2/3}}{\alpha^{4/3}} + \frac{\sqrt{k}}{\alpha\sqrt{\varepsilon}}\right)$ |
| | **ELSE** |
| | $\Omega\left(\frac{\sqrt{k}}{\alpha^2} + \frac{\sqrt{k}}{\alpha\sqrt{\varepsilon}} + \frac{1}{\alpha\varepsilon}\right) \leq S(\text{CT}, k, \alpha, \varepsilon) \leq O\left(\frac{\sqrt{k}}{\alpha^2} + \frac{1}{\alpha^2\varepsilon}\right)$ [Theorem 4] |

Table 1: Summary of the sample complexity bounds for $\varepsilon$-DP identity, and closeness testing. For $(\varepsilon, \delta)$-DP algorithms, we can simply replace $\varepsilon$ in the sample complexity by $(\varepsilon + \delta)$.

## 1.2 Related Work

A number of papers have recently studied hypothesis testing problems under differential privacy guarantees [39, 40, 41]. Some works analyze the distribution of the test statistic in the asymptotic regime. The work most closely related to ours is [34], which studied identity testing in the finite sample regime. We mentioned their guarantees along with our results on identity testing in the previous section.

There has been a line of research for statistical testing and estimation problems under the notion of *local* differential privacy [24, 23, 42, 43, 44, 45, 46, 47, 48, 49]. These papers study some basic statistical problems and provide minimax lower bounds using Fano's inequality. [50] studies structured distribution estimation under differential privacy. Information theoretic approaches to data privacy have been studied recently using quantities like mutual information, and guessing probability to quantify privacy [51, 52, 53, 54, 55].

[56, 57] provide methods to prove lower bounds on DP algorithms via packing. Recently, [58] use coupling to prove lower bounds on the sample complexity for differentially private confidence intervals. Our results are more general, in that, we can handle mixtures of distributions, which can provide optimal lower bounds on identity testing. [59, 60] characterize

differential privacy through a coupling argument. [61] also uses the idea of coupling implicitly when designing differentially private partition algorithms. [62] uses our coupling argument to prove lower bounds for differentially private property estimation problems.

In a contemporaneous and independent work, [63], the authors study the same problems that we consider, and obtain the same upper bounds for the sparse case, when $m \leq k$. They also provide experimental results to show the performance of the privatized algorithms. However, their results are sub-optimal for $m = \Omega(k)$ for identity testing, and they do not provide any lower bounds for the problems. Both [34], and [63] consider only pure-differential privacy, which are a special case of our results.

**Organization of the paper.** In Section 2, we discuss the definitions and notations. A general technique for proving lower bounds for differentially private algorithms is described in Section 3. Section 4 gives upper and lower bounds for identity testing, and closeness testing is studied in Section 5.

## 2 Preliminaries

Let $\Delta_k$ be the class of all discrete distributions over a domain of size $k$, which wlog is assumed to be $[k] := \{1, \ldots, k\}$. We denote length-$m$ samples $X_1, \ldots, X_m$ by $X_1^m$. For $x \in [k]$, let $p_x$ be the probability of $x$ under $p$. Let $M_x(X_1^m)$ be the number of times $x$ appears in $X_1^m$. For $A \subseteq [k]$, let $p(A) = \sum_{x \in A} p_x$. Let $X \sim p$ denote that the random variable $X$ has distribution $p$. Let $u[k]$ be the uniform distribution over $[k]$, and $B(b)$ be the Bernoulli distribution with bias $b$. The *total variation* distance between distributions $p$, and $q$ over $[k]$ is $d_{TV}(p, q) := \sup_{A \subset [k]} \{p(A) - q(A)\} = \frac{1}{2} \|p - q\|_1$.

**Definition 1.** Let $p$, and $q$ be distributions over $\mathcal{X}$, and $\mathcal{Y}$ respectively. A *coupling* between $p$ and $q$ is a distribution over $\mathcal{X} \times \mathcal{Y}$ whose marginals are $p$ and $q$ respectively.

**Definition 2.** The *Hamming distance* between two sequences $X_1^m$ and $Y_1^m$ is $d_H(X_1^m, Y_1^m) := \sum_{i=1}^m \mathbb{I}\{X_i \neq Y_i\}$, the number of positions where $X_1^m$, and $Y_1^m$ differ.

**Definition 3.** A randomized algorithm $\mathcal{A}$ on a set $\mathcal{X}^m \to \mathcal{S}$ is said to be $(\varepsilon, \delta)$-differentially private if for any $S \subset \mathrm{range}(\mathcal{A})$, and all pairs of $X_1^m$, and $Y_1^m$ with $d_H(X_1^m, Y_1^m) \leq 1$ such that $\Pr\left(\mathcal{A}(X_1^m) \in S\right) \leq e^\varepsilon \cdot \Pr\left(\mathcal{A}(Y_1^m) \in S\right) + \delta$.

The case when $\delta = 0$ is called *pure differential privacy*. For simplicity, we denote pure differential privacy as $\varepsilon$-differential privacy ($\varepsilon$-DP).

Next we state the group property of differential privacy. We give a proof in Appendix A.1.

**Lemma 1.** *Let $\mathcal{A}$ be a $(\varepsilon, \delta)$-DP algorithm, then for sequences $x_1^m$, and $y_1^m$ with $d_H(x_1^m, y_1^m) \leq t$, and $\forall S \subset \mathrm{range}(\mathcal{A})$, $\Pr\left(\mathcal{A}(x_1^m) \in S\right) \leq e^{t\varepsilon} \cdot \Pr\left(\mathcal{A}(y_1^m) \in S\right) + \delta t e^{\varepsilon(t-1)}$.*

The next two lemmas state a relationship between $(\varepsilon, \delta)$ and $\varepsilon$-differential privacy. We give a proof of Lemma 2 in Appendix A.2. And Lemma 3 follows from [34].

**Lemma 2.** *Any $(\varepsilon + \delta, 0)$- differentially private algorithm is also $(\varepsilon, \delta)$-differentially private.*

**Lemma 3.** *An $(\varepsilon, \delta)$-DP algorithm for a testing problem can be converted to an $(\varepsilon + c\delta, 0)$ algorithm for some constant $c > 0$.*

Combining these two results, it suffices to prove bounds for $(\varepsilon, 0)$-DP, and plug in $\varepsilon$ with $(\varepsilon + \delta)$ to obtain bounds that are tight up to constant factors for $(\varepsilon, \delta)$-DP.

The notion of sensitivity is useful in establishing bounds under differential privacy.

**Definition 4.** The *sensitivity* of $f : [k]^m \to \mathbb{R}$ is

$$\Delta(f) := \max_{d_H(X_1^m, Y_1^m) \leq 1} \left|f(X_1^m) - f(Y_1^m)\right|.$$

For $x \in \mathbb{R}$, $\sigma(x) := \frac{1}{1 + \exp(-x)} = \frac{\exp(x)}{1 + \exp(x)}$ is the sigmoid function. The following properties follow from the definition of $\sigma$.

**Lemma 4.**        *1. For all $x, \gamma \in \mathbb{R}$, $\exp(-|\gamma|) \leq \frac{\sigma(x+\gamma)}{\sigma(x)} \leq \exp(|\gamma|)$.*

   *2. Let $0 < \eta < \frac{1}{2}$. Suppose $x \geq \log \frac{1}{\eta}$. Then $\sigma(x) > 1 - \eta$.*

**Identity Testing (IT).** Given description of $q \in \Delta_k$ over $[k]$, parameters $\alpha$, and $m$ independent samples $X_1^m$ from unknown $p \in \Delta_k$. $\mathcal{A}$ is an $(k, \alpha)$-identity testing algorithm for $q$, if when $p = q$, $\mathcal{A}$ outputs "$p = q$" with probability at least 0.9, and when $d_{TV}(p, q) \geq \alpha$, $\mathcal{A}$ outputs "$p \neq q$" with probability at least 0.9.

**Definition 5.** The sample complexity of DP-identity testing, denoted $S(\mathtt{IT}, k, \alpha, \varepsilon)$, is the smallest $m$ for which there exists an $\varepsilon$-DP algorithm $\mathcal{A}$ that uses $m$ samples to achieve $(k, \alpha)$-identity testing. Without privacy concerns, $S(\mathtt{IT}, k, \alpha)$ denotes the sample complexity. When $q = u[k]$, the problem reduces to uniformity testing, and the sample complexity is denoted as $S(\mathtt{UT}, k, \alpha, \varepsilon)$.

**Closeness Testing (CT).** Given $m$ independent samples $X_1^m$, and $Y_1^m$ from unknown distributions $p$, and $q$. An algorithm $\mathcal{A}$ is an $(k, \alpha)$-closeness testing algorithm if when $p = q$, $\mathcal{A}$ outputs $p = q$ with probability at least 0.9, and when $d_{TV}(p, q) \geq \alpha$, $\mathcal{A}$ outputs $p \neq q$ with probability at least 0.9.

**Definition 6.** The sample complexity of DP-closeness testing, denoted $S(\mathtt{CT}, k, \alpha, \varepsilon)$, is the smallest $m$ for which there exists an $\varepsilon$-DP algorithm $\mathcal{A}$ that uses $m$ samples to achieve $(k, \alpha)$-closeness testing. When privacy is not a concern, we denote the sample complexity of closeness testing as $S(\mathtt{CT}, k, \alpha)$.

**Hypothesis Testing (HT).** Suppose we have distributions $p$ and $q$ over $\mathcal{X}^m$, and $X_1^m \sim p, Y_1^m \sim q$, we say an algorithm $\mathcal{A} : \mathcal{X}^m \to \{p, q\}$ can distinguish between $p$ and $q$ if $\Pr\left(\mathcal{A}(X_1^m) = q\right) < 0.1$ and $\Pr\left(\mathcal{A}(Y_1^m) = p\right) < 0.1$.

## 3    Privacy Bounds Via Coupling

Recall that *coupling* between distributions $p$ and $q$ over $\mathcal{X}$, and $\mathcal{Y}$, is a distribution over $\mathcal{X} \times \mathcal{Y}$ whose marginal distributions are $p$ and $q$ (Definition 1). For simplicity, we treat coupling as a randomized function $f : \mathcal{X} \to \mathcal{Y}$ such that if $X \sim p$, then $Y = f(X) \sim q$. Note that $X$, and $Y$ are not necessarily independent.

*Example* 1. Let $B(b_1)$, and $B(b_2)$ be Bernoulli distributions with bias $b_1$, and $b_2$ such that $b_1 < b_2$. Let $p$, and $q$ be distributions over $\{0,1\}^m$ obtained by $m$ *i.i.d.* samples from $B(b_1)$, and $B(b_2)$ respectively. Let $X_1^m$ be distributed according to $p$. Generate a sequence $Y_1^m$ as follows: If $X_i = 1$, then $Y_i = 1$. If $X_i = 0$, we flip another coin with bias $(b_2 - b_1)/(1 - b_1)$, and let $Y_i$ be the output of this coin. Repeat the process independently for each $i$, such that the $Y_i$'s are all independent of each other. Then $\Pr\left(Y_i = 1\right) = b_1 + (1 - b_1)(b_2 - b_1)/(1 - b_1) = b_2$, and $Y_1^m$ is distributed according to $q$.

We would like to use coupling to prove lower bounds on differentially private algorithms for testing problems. Let $p$ and $q$ be distributions over $\mathcal{X}^m$. If there is a coupling between $p$ and $q$ with a small *expected* Hamming distance, we might expect that the algorithm cannot have strong privacy guarantees. The following theorem formalizes this intuition:

**Theorem 1.** *Suppose there is a coupling between $p$ and $q$ over $\mathcal{X}^m$, such that $\mathbb{E}\left[d_H(X_1^m, Y_1^m)\right] \leq D$ where $X_1^m \sim p, Y_1^m \sim q$. Then, any $(\varepsilon, \delta)$-differentially private hypothesis testing algorithm $\mathcal{A} : \mathcal{X}^m \to \{p, q\}$ on $p$ and $q$ must satisfy $\varepsilon + \delta = \Omega\left(\frac{1}{D}\right)$*

*Proof.* Let $(X_1^m, Y_1^m)$ be distributed according to a coupling of $p$, and $q$ with $\mathbb{E}\left[d_H(X_1^m, Y_1^m)\right] \leq D$. By Markov's inequality, $\Pr\left(d_H(X_1^m, Y_1^m) > 10D\right) < \Pr\left(d_H(X_1^m, Y_1^m) > 10 \cdot \mathbb{E}\left[d_H(X_1^m, Y_1^m)\right]\right) < 0.1$. Let $x_1^m$ and $y_1^m$ be the realization of $X_1^m$ and $Y_1^m$. Let $W = \{(x_1^m, y_1^m) | d_H(x_1^m, y_1^m) \leq 10D\}$. Then we have

$$0.1 \geq \Pr\left(\mathcal{A}(X_1^m) = q\right) \geq \sum_{(x_1^m, y_1^m) \in W} \Pr\left(X_1^m = x_1^m, Y_1^m = y_1^m\right) \cdot \Pr\left(\mathcal{A}(x_1^m) = q\right).$$

By Lemma 1, and $\Pr\left(d_H(X_1^m, Y_1^m) > 10D\right) < 0.1$, and $\Pr\left(\mathcal{A}(y_1^m) = q\right) \leq 1$,

$$\Pr\left(\mathcal{A}(Y_1^m) = q\right) \leq \sum_{(x_1^m, y_1^m) \in W} \Pr\left(x_1^m, y_1^m\right) \cdot \Pr\left(\mathcal{A}(y_1^m) = q\right) + \sum_{(x_1^m, y_1^m) \notin W} \Pr\left(x_1^m, y_1^m\right) \cdot 1$$

$$\leq \sum_{(x_1^m, y_1^m) \in W} \Pr\left(x_1^m, y_1^m\right) \cdot \left(e^{\varepsilon \cdot 10D} \Pr\left(\mathcal{A}(x_1^m) = q\right) + 10D\delta \cdot e^{\varepsilon \cdot 10(D-1)}\right) + 0.1$$

$$\leq 0.1 e^{\varepsilon \cdot 10D} + 10D\delta \cdot e^{\varepsilon \cdot 10D} + 0.1.$$

Since we know $\Pr\left(\mathcal{A}(Y_1^m) = q\right) > 0.9$, then $0.9 < \Pr\left(\mathcal{A}(Y_1^m) = q\right) < 0.1 e^{\varepsilon \cdot 10D} + 10D\delta \cdot e^{\varepsilon \cdot 10D} + 0.1$. Hence, either $e^{\varepsilon \cdot 10D} = \Omega(1)$ or $10D\delta = \Omega(1)$, which implies that $D = \Omega\left(\min\left\{\frac{1}{\varepsilon}, \frac{1}{\delta}\right\}\right) = \Omega\left(\frac{1}{\varepsilon + \delta}\right)$, proving the theorem. $\square$

Set $\delta = 0$, we obtain the bound for pure differential privacy. In the next few sections, we use this theorem to get sample complexity bounds for differentially private testing problems.

## 4 Identity Testing

In this section, we prove the bounds for identity testing. Our main result is the following.

**Theorem 2.**

$$S(\mathtt{IT}, k, \alpha, \varepsilon) = \Theta\left(\frac{k^{1/2}}{\alpha^2} + \max\left\{\frac{k^{1/2}}{\alpha \varepsilon^{1/2}}, \frac{k^{1/3}}{\alpha^{4/3} \varepsilon^{2/3}}, \frac{1}{\alpha \varepsilon}\right\}\right).$$

*Or we can write it according to the parameter range,*

$$S(\mathtt{IT}, k, \alpha, \varepsilon) = \begin{cases} \Theta\left(\frac{\sqrt{k}}{\alpha^2} + \frac{k^{1/2}}{\alpha \varepsilon^{1/2}}\right), & \text{when } k = \Omega\left(\frac{1}{\alpha^4}\right) \text{ and } k = \Omega\left(\frac{1}{\alpha^2 \varepsilon}\right), \\ \Theta\left(\frac{\sqrt{k}}{\alpha^2} + \frac{k^{1/3}}{\alpha^{4/3} \varepsilon^{2/3}}\right), & \text{when } k = \Omega\left(\frac{\alpha}{\varepsilon}\right) \text{ and } k = O\left(\frac{1}{\alpha^4} + \frac{1}{\alpha^2 \varepsilon}\right), \\ \Theta\left(\frac{\sqrt{k}}{\alpha^2} + \frac{1}{\alpha \varepsilon}\right), & \text{when } k = O\left(\frac{\alpha}{\varepsilon}\right). \end{cases}$$

Our bounds are tight up to constant factors in all parameters. To get the sample complexity for $(\varepsilon, \delta)$-differential privacy, we can simply replace $\varepsilon$ by $(\varepsilon + \delta)$.

In Theorem 3 we will show a reduction from identity to uniformity testing under pure differential privacy. Using this, it will be enough to design algorithms for uniformity testing, which is done in Section 4.2.

Moreover since uniformity testing is a special case of identity testing, any lower bound for uniformity will port over to identity, and we give such bounds in Section 4.3.

### 4.1 Uniformity Testing implies Identity Testing

The sample complexity of testing identity of any distribution is $O(\frac{\sqrt{k}}{\alpha^2})$, a bound that is tight for the uniform distribution. Recently [35] proposed a scheme to reduce the problem of testing identity of distributions over $[k]$ for total variation distance $\alpha$ to the problem of testing uniformity over $[6k]$ with total variation parameter $\alpha/3$. In other words, they show that $S(\mathtt{IT}, k, \alpha) \leq S(\mathtt{UT}, 6k, \alpha/3)$. Building on [35], we prove that a similar bound also holds for differentially private algorithms. The proof is in Appendix B.

**Theorem 3.** $S(\mathtt{IT}, k, \alpha, \varepsilon) \leq S(\mathtt{UT}, 6k, \alpha/3, \varepsilon)$.

### 4.2 Identity Testing – Upper Bounds

In this section, we will show that by privatizing the statistic proposed in [38] we can achieve the sample complexity in Theorem 2 for all parameter ranges. The procedure is described in Algorithm 1.

Recall that $M_x(X_1^m)$ is the number of appearances of $x$ in $X_1^m$. Let

$$S(X_1^m) := \frac{1}{2} \cdot \sum_{x=1}^{n} \left| \frac{M_x(X_1^m)}{m} - \frac{1}{k} \right|, \tag{1}$$

be the TV distance from the empirical distribution to the uniform distribution. Let $\mu(p) = \mathbb{E}\left[S(X_1^m)\right]$ when the samples are drawn from distribution $p$. They show the following separation result on the expected value of $S(X_1^m)$.

**Lemma 5** ([38]). *Let $p$ be a distribution over $[k]$ and $d_{TV}(p, u[k]) \geq \alpha$, then there is a constant $c$ such that*

$$\mu(p) - \mu(u[k]) \geq c\alpha^2 \min\left\{ \frac{m^2}{k^2}, \sqrt{\frac{m}{k}}, \frac{1}{\alpha} \right\}.$$

[38] used this result to show that thresholding $S(X_1^m)$ at 0 is an optimal algorithm for identity testing. We first normalize the statistic to simplify the presentation of our DP algorithm. Let

$$Z(X_1^m) := \begin{cases} k\left(S(X_1^m) - \mu(u[k]) - \frac{1}{2}c\alpha^2 \cdot \frac{m^2}{k^2}\right), & \text{when } m \leq k, \\ m\left(S(X_1^m) - \mu(u[k]) - \frac{1}{2}c\alpha^2 \cdot \sqrt{\frac{m}{k}}\right), & \text{when } k < m \leq \frac{k}{\alpha^2}, \\ m\left(S(X_1^m) - \mu(u[k]) - \frac{1}{2}c\alpha\right), & \text{when } m \geq \frac{k}{\alpha^2}. \end{cases} \tag{2}$$

where $c$ is the constant in Lemma 5, and $\mu(u[k])$ is the expected value of $S(X_1^m)$ when $X_1^m$ are drawn from uniform distribution.

---

**Algorithm 1** Uniformity testing

**Input:** $\varepsilon$, $\alpha$, i.i.d. samples $X_1^m$ from $p$
1: Let $Z(X_1^m)$ be evaluated from (1), and (2).
2: Generate $Y \sim B(\sigma(\varepsilon \cdot Z))$, $\sigma$ is the sigmoid function.
3: **if** $Y = 0$, **return** $p = u[k]$, **else, return** $p \neq u[k]$.

---

We now prove that this algorithm is $\varepsilon$-DP. We need the following sensitivity result.

**Lemma 6.** $\Delta(Z) \leq 1$ *for all values of $m$, and $k$.*

*Proof.* Recall that $S(X_1^m) = \frac{1}{2} \cdot \sum_{x=1}^{n} \left| \frac{M_x(X_1^m)}{m} - \frac{1}{k} \right|$. Changing any one symbol changes at most two of the $M_x(X_1^m)$'s. Therefore at most two of the terms change by at most $\frac{1}{m}$. Therefore, $\Delta(S(X_1^m)) \leq \frac{1}{m}$, for any $m$. When $m \leq k$, this can be strengthened with observation that $M_x(X_1^m)/m \geq \frac{1}{k}$, for all $M_x(X_1^m) \geq 1$. Therefore, $S(X_1^m) = \frac{1}{2} \cdot \left(\sum_{x:M_x(X_1^m)\geq 1} \left(\frac{M_x(X_1^m)}{m} - \frac{1}{k}\right) + \sum_{x:M_x(X_1^m)=0} \frac{1}{k}\right) = \frac{\Phi_0(X_1^m)}{k}$, where $\Phi_0(X_1^m)$ is the number of symbols not appearing in $X_1^m$. This changes by at most one when one symbol is changed, proving the result. $\square$

Using this lemma, $\varepsilon \cdot Z(X_1^m)$ changes by at most $\varepsilon$ when $X_1^m$ is changed at one location. Invoking Lemma 4, the probability of any output changes by a multiplicative $\exp(\varepsilon)$, and the algorithm is $\varepsilon$-differentially private.

To prove the sample complexity bound, we first show that the mean of the test statistic is well separated using Lemma 5. Then we use the concentration bound of the test statistic from [38] to get the final complexity. Due to lack of space, the detailed proof of sample complexity bound is given in Appendix C.

## 4.3 Sample Complexity Lower bounds for Uniformity Testing

In this section, we will show the lower bound part of Theorem 2. The first term is the lower bound without privacy constraints, proved in [7]. In this section, we will prove the terms associated with privacy.

The simplest argument is for $m \geq \frac{k}{\alpha^2}$, which hopefully will give you a sense of how coupling argument works. We consider the case of binary identity testing where the goal is to test whether the bias of a coin is $1/2$ or $\alpha$-far from $1/2$. This is a special case of identity testing for distributions over $[k]$ (when $k-2$ symbols have probability zero). This is strictly harder than the problem of distinguishing between $B(1/2)$ and $B(1/2 + \alpha)$. The coupling given in Example 1 has expected hamming distance of $\alpha m$. Hence combing with Theorem 1, we get a lower bound of $\Omega(\frac{1}{\alpha \varepsilon})$.

We now consider the cases $m \leq k$ and $k < m \leq \frac{k}{\alpha^2}$.

To this end, we invoke LeCam's two point theorem, and design a hypothesis testing problem that will imply a lower bound on uniformity testing. The testing problem will be to distinguish between the following two cases.

**Case 1:** We are given $m$ independent samples from the uniform distribution $u[k]$.

**Case 2:** Generate a distribution $p$ with $d_{TV}(p, u[k]) \geq \alpha$ according to some prior over all such distributions. We are then given $m$ independent samples from this distribution $p$.

Le Cam's two point theorem [64] states that any lower bound for distinguishing between these two cases is a lower bound on identity testing problem.

We now describe the prior construction for **Case 2**, which is the same as considered by [7] for lower bounds on identity testing without privacy considerations. For each $\mathbf{z} \in \{\pm 1\}^{k/2}$, define a distribution $p_{\mathbf{z}}$ over $[k]$ such that

$$p_{\mathbf{z}}(2i-1) = \frac{1 + \mathbf{z}_i \cdot 2\alpha}{k}, \text{ and } p_{\mathbf{z}}(2i) = \frac{1 - \mathbf{z}_i \cdot 2\alpha}{k}.$$

Then for any $\mathbf{z}$, $d_{TV}(P_{\mathbf{z}}, u[k]) = \alpha$. For **Case 2**, choose $p$ uniformly from these $2^{k/2}$ distributions. Let $Q_2$ denote the distribution on $[k]^m$ by this process. In other words, $Q_2$ is a mixture of product distributions over $[k]$.

In **Case 1**, let $Q_1$ be the distribution of $m$ *i.i.d.* samples from $u[k]$.

To obtain a sample complexity lower bound for distinguishing the two cases, we will design a coupling between $Q_1$, and $Q_2$, and bound its expected Hamming distance. While it can be shown that the Hamming distance of the coupling between the uniform distribution with any *one* of the $2^{k/2}$ distributions grows as $\alpha m$, it can be significantly smaller, when we consider the mixtures. In particular, the following lemma shows that there exist couplings with bounded Hamming distance.

**Lemma 7.** *There is a coupling between $X_1^m$ generated by $Q_1$, and $Y_1^m$ by $Q_2$ such that*

$$\mathbb{E}\left[d_H(X_1^m, Y_1^m)\right] \leq C \cdot \alpha^2 \min\{\tfrac{m^2}{k}, \tfrac{m^{3/2}}{k^{1/2}}\}.$$

The lemma is proved in Appendix D. Now applying Theorem 1, we get the bound in Theorem 2.

## 5    Closeness Testing

Recall the closeness testing problem from Section 2, and the tight non-private bounds from Table 1. Our main result in this section is the following theorem characterizing the sample complexity of differentially private algorithms for closeness testing.

**Theorem 4.** *If $\alpha > 1/k^{1/4}$, and $\varepsilon \alpha^2 > 1/k$,*

$$S(\mathtt{CT}, k, \alpha, \varepsilon) = \Theta\left(\frac{k^{2/3}}{\alpha^{4/3}} + \frac{k^{1/2}}{\alpha\sqrt{\varepsilon}}\right),$$

*otherwise,*

$$\Omega\left(\frac{k^{1/2}}{\alpha^2} + \frac{k^{1/2}}{\alpha\sqrt{\varepsilon}} + \frac{1}{\alpha\varepsilon}\right) \leq S(\mathtt{CT}, k, \alpha, \varepsilon) \leq O\left(\frac{k^{1/2}}{\alpha^2} + \frac{1}{\alpha^2\varepsilon}\right).$$

This theorem shows that in the sparse regime, when $m = O(k)$, our bounds are tight up to constant factors in all parameters. To prove the upper bounds, we only consider the case when $\delta = 0$, which would suffice by lemma 2. We privatize the closeness testing algorithm of [10]. To reduce the strain on the readers, we drop the sequence notations explicitly and let

$$\mu_i := M_i(X_1^m), \text{ and } \nu_i := M_i(Y_1^m).$$

The statistic used by [10] is

$$Z(X_1^m, Y_1^m) := \sum_{i \in [k]} \frac{(\mu_i - \nu_i)^2 - \mu_i - \nu_i}{\mu_i + \nu_i},$$

where we assume that $((\mu_i - \nu_i)^2 - \mu_i - \nu_i)/(\mu_i + \nu_i) = 0$, when $\mu_i + \nu_i = 0$. It turns out that this statistic has a constant sensitivity, as shown in Lemma 8.

**Lemma 8.** $\Delta(Z(X_1^m, Y_1^m)) \leq 14$.

*Proof.* Since $Z(X_1^m, Y_1^m)$ is symmetric, without loss of generality assume that one of the symbols is changed in $Y_1^m$. This would cause at most two of the $\nu_i$'s to change. Suppose $\nu_i \geq 1$, and it changed to $\nu_i - 1$. Suppose, $\mu_i + \nu_i > 1$, the absolute change in the $i$th term of the statistic is

$$
\left| \frac{(\mu_i - \nu_i)^2}{\mu_i + \nu_i} - \frac{(\mu_i - \nu_i + 1)^2}{\mu_i + \nu_i - 1} \right| = \left| \frac{(\mu_i + \nu_i)(2\mu_i - 2\nu_i + 1) + (\mu_i - \nu_i)^2}{(\mu_i + \nu_i)(\mu_i + \nu_i - 1)} \right|
$$
$$
\leq \left| \frac{2\mu_i - 2\nu_i + 1}{\mu_i + \nu_i - 1} \right| + \left| \frac{\mu_i - \nu_i}{\mu_i + \nu_i - 1} \right|
$$
$$
\leq \frac{3|\mu_i - \nu_i| + 1}{\mu_i + \nu_i - 1} \leq 3 + \frac{4}{\mu_i + \nu_i - 1} \leq 7.
$$

When $\mu_i + \nu_i = 1$, the change can again be bounded by 7. Since at most two of the $\nu_i$'s change, we obtain the desired bound. □

We use the same approach with the test statistic as with uniformity testing to obtain a differentially private closeness testing method, described in Algorithm 2. Since the sensitivity of the statistic is at most 14, the input to the sigmoid changes by at most $\varepsilon$ when any input sample is changed. Invoking Lemma 4, the probability of any output changes by a multiplicative $\exp(\varepsilon)$, and the algorithm is $\varepsilon$-differentially private.

---
**Algorithm 2**
___
    **Input:** $\varepsilon$, $\alpha$, sample access to distribution $p$ and $q$

1: $Z' \leftarrow (Z(X_1^m, Y_1^m) - \frac{1}{2}\frac{m^2\alpha^2}{4k+2m})/14$
2: Generate $Y \sim B(\sigma(\exp(\varepsilon \cdot Z'))$
3:  **if** $Y = 0$, **return** $p = q$
4:  **else**, **return** $p \neq q$

---

The remaining part is to show that Algorithm 2 satisfies sample complexity upper bounds described in theorem 4. We will give the details in Appendix E, where the analysis of the lower bound is also given.

## Acknowledgement

The authors thank Gautam Kamath for some very helpful suggestions about this work.

## Footnotes

*The authors are listed in alphabetical order. This research was supported by NSF-CCF-CRII 1657471, and a grant from Cornell University.

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
