[Supplementary Material]

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

(\mathcal{A}(X_1^m) \in S) \leq e^\varepsilon \cdot \Pr(\mathcal{A}(Y_1^m) \in S) + \delta$.

The case when $\delta = 0$ is called *pure differential privacy*. For simplicity, we denote pure differential privacy as $\varepsilon$-differential privacy ($\varepsilon$-DP).

Next we state the group property of differential privacy. We give a proof in Appendix A.1.

**Lemma 1.** *Let $\mathcal{A}$ be a $(\varepsilon, \delta)$-DP algorithm, then for sequences $x_1^m$, and $y_1^m$ with $d_H(x_1^m, y_1^m) \leq t$, and $\forall S \subset range(\mathcal{A})$, $\Pr(\mathcal{A}(x_1^m) \in S) \leq e^{t\varepsilon} \cdot \Pr(\mathcal{A}(y_1^m) \in S) + \

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

# A  Proof of Lemmas

## A.1  Proof of Lemma 1

*Proof.* Let $d_H(x_1^m, y_1^m) = \hat{D}$. When $\hat{D} = 0$ or 1, the lemma is trivially true. Then, suppose $\hat{D} \geq 2$ we can find $\hat{D} - 1$ sequences $z_1^m, \ldots, z_{\hat{D}-1}^m$ over $\mathcal{X}^m$ with $d_H(x_1^m, z_1^m) = 1, d_H(z_{\hat{D}-1}^m, y_1^m) = 1$ and $d_H(z_i^m, z_{i+1}^m) = 1$ for $i \in \{1, 2, ..., \hat{D} - 2\}$. Hence, by the condition of $(\varepsilon, \delta)$ - differential privacy,

$$\Pr\left(\mathcal{A}(x_1^m) = q\right) \leq e^\varepsilon \Pr\left(\mathcal{A}(z_1^m) = q\right) + \delta \leq e^\varepsilon (e^\varepsilon \Pr\left(\mathcal{A}(z_2^m) = q\right) + \delta) + \delta \leq \ldots$$

$$\leq e^{\hat{D}\varepsilon} \Pr\left(\mathcal{A}(y_1^m) = q\right) + \delta \cdot \sum_{i=0}^{\hat{D}-1} e^{i\varepsilon} \leq e^{\hat{D}\varepsilon} \Pr\left(\mathcal{A}(y_1^m) = q\right) + \delta\hat{D}e^{(\hat{D}-1)\varepsilon}$$

$$\leq e^{t\varepsilon} \Pr\left(\mathcal{A}(y_1^m) = q\right) + \delta t e^{\varepsilon(t-1)}.$$

$\square$

## A.2  Proof of Lemma 2

*Proof.* Suppose $\mathcal{A}$ is a $(\varepsilon + \delta)$-differentially private algorithm. Then for any $X_1^m$ and $Y_1^m$ with $d_H(X_1^m, Y_1^m) \leq 1$ and any $S \subset \text{range}(\mathcal{A})$, we have

$$\Pr\left(\mathcal{A}(X_1^m) \in S\right) \leq e^\varepsilon \cdot \Pr\left(\mathcal{A}(Y_1^m) \in S\right) + (e^\delta - 1) \cdot e^\varepsilon \Pr\left(\mathcal{A}(Y_1^m) \in S\right).$$

If $e^\varepsilon \cdot \Pr\left(\mathcal{A}(Y_1^m) \in S\right) > 1 - \delta$, then $\Pr\left(\mathcal{A}(X_1^m) \in S\right) \leq 1 < e^\varepsilon \cdot \Pr\left(\mathcal{A}(Y_1^m) \in S\right) + \delta$. Otherwise, $e^\varepsilon \cdot \Pr\left(\mathcal{A}(Y_1^m) \in S\right) \leq 1 - \delta$. To prove $(e^\delta - 1) \cdot e^\varepsilon \cdot \Pr\left(\mathcal{A}(Y_1^m) \in S\right) < \delta$, it suffices to show $(e^\delta - 1)(1 - \delta) \leq \delta$, which is equivalent to $e^{-\delta} \geq 1 - \delta$, completing the proof. $\square$

# B  Proof of Theorem 3

*Proof.* We first briefly describe the essential components of the construction of [35]. Given an explicit distribution $q$ over $[k]$, there exists a randomized function $F_q : [k] \to [6k]$ such that if $X \sim q$, then $F_q(X) \sim u[6k]$, and if $X \sim p$ for a distribution with $d_{TV}(p, q) \geq \alpha$, then the distribution of $F_q(X)$ has a total variation distance of at least $\alpha/3$ from $u[6k]$. Given $s$ samples $X_1^s$ from a distribution $p$ over $[k]$. Apply $F_q$ independently to each of the $X_i$ to obtain a new sequence $Y_1^s = F_q(X_1^s) := F_q(X_1) \ldots F_q(X_s)$. Let $\mathcal{A}$ be an algorithm that distinguishes $u[6k]$ from all distributions with total variation distance at least $\alpha/3$ from it. Then consider the algorithm $\mathcal{A}'$ that outputs $p = q$ if $\mathcal{A}$ outputs "$p = u[6k]$", and outputs $p \neq q$ otherwise. This shows that without privacy constraints, $S(\text{IT}, k, \alpha) \leq S(\text{UT}, 6k, \alpha/3)$ (See [35] for details).

We now prove that if further $\mathcal{A}$ was an $\varepsilon$-DP algorithm, then $\mathcal{A}'$ is also an $\varepsilon$-DP algorithm. Suppose $X_1^s$, and $X_1'^s$ be two sequences in $[k]^s$ that could differ only on the last coordinate, namely $X_1^s = X_1^{s-1}X_s$, and $X_1'^s = X_1^{s-1}X_s'$.

Consider two sequences $Y_1^s = Y_1^{s-1}Y_s$, and $Y_1'^s = Y_1^{s-1}Y_s'$ in $[6k]^s$ that could differ on only the last coordinate. Since $\mathcal{A}$ is $\varepsilon$-DP,

$$\mathcal{A}(Y_1^s = u[6k]) \leq \mathcal{A}(Y_1'^s = u[6k]) \cdot e^\varepsilon. \tag{3}$$

Moreover, since $F_q$ is applied independently to each coordinate,

$$\Pr\left(F_q(X_1^s) = Y_1^s\right) = \Pr\left(F_q(X_1^{s-1}) = Y_1^{s-1}\right) \Pr\left(F_q(X_s) = Y_s\right).$$

Then,

$$\Pr\left(\mathcal{A}^{'}(X_1^s) = q\right)$$

$$= \Pr\left(\mathcal{A}(F_q(X_1^s)) = u[6k]\right)$$

$$= \sum_{Y_1^s} \Pr\left(\mathcal{A}(Y_1^s) = u[6k]\right) \Pr\left(F_q(X_1^s) = Y_1^s\right)$$

$$= \sum_{Y_1^{s-1}} \sum_{Y_s \in [6k]} \Pr\left(\mathcal{A}(Y_1^s) = u[6k]\right) \Pr\left(F_q(X_1^{s-1}) = Y_1^{s-1}\right) \Pr\left(F_q(X_s) = Y_s\right)$$

$$= \sum_{Y_1^{s-1}} \Pr\left(F_q(X_1^{s-1}) = Y_1^{s-1}\right) \left[ \sum_{Y_s \in [6k]} \Pr\left(\mathcal{A}(Y_1^s) = u[6k]\right) \Pr\left(F_q(X_s) = Y_s\right) \right]. \quad (4)$$

Similarly,

$$\Pr\left(\mathcal{A}^{'}(X_1^{'s}) = q\right) = \sum_{Y_1^{s-1}} \Pr\left(F_q(X_1^{s-1}) = Y_1^{s-1}\right) \left[ \sum_{Y_s' \in [6k]} \Pr\left(\mathcal{A}(Y_1^{'s}) = u[6k]\right) \Pr\left(F_q(X_s') = Y_s'\right) \right]. \quad (5)$$

For a fixed $Y_1^{s-1}$, the term within the bracket in (4), and (5) are both expectations over the final coordinate. However, by (3) these expectations differ at most by a multiplicative $e^\varepsilon$ factor. This implies that

$$\Pr\left(\mathcal{A}'(X_1^s) = q\right) \le \Pr\left(\mathcal{A}'(X_1^{'s}) = q\right) e^\varepsilon.$$

The argument is similar for the case when the testing output is **not** $u[6k]$, and is omitted here. We only considered sequences that differ on the last coordinate, and the proof remains the same when any of the coordinates is changed. This proves the privacy guarantees of the algorithm. □

## C  Sample Complexity Bound of Algorithm 1

In this section, we prove the sample complexity bound of Algorithm 1 where we privatize the statistic proposed in [38] to achieve the sample complexity in Theorem 2 for all parameter ranges.

Because of the normalization in Equation 2 and lemma 5, for $X_1^m$ drawn from $u[k]$

$$\mathbb{E}\left[Z(X_1^m)\right] \le \begin{cases} -\frac{1}{2}c\alpha^2 \cdot \frac{m^2}{k}, & \text{when } m \le k, \\ -\frac{1}{2}c\alpha^2 \cdot \frac{m^{3/2}}{k^{1/2}}, & \text{when } k < m \le \frac{k}{\alpha^2}, \\ -\frac{1}{2}cm\alpha, & \text{when } m \ge \frac{k}{\alpha^2} . \end{cases} \quad (6)$$

For $X_1^m$ drawn from $p$ with $d_{TV}(p, u[k]) \ge \alpha$,

$$\mathbb{E}\left[Z(X_1^m)\right] \ge \begin{cases} \frac{1}{2}c\alpha^2 \cdot \frac{m^2}{k}, & \text{when } m \le k, \\ \frac{1}{2}c\alpha^2 \cdot \frac{m^{3/2}}{k^{1/2}}, & \text{when } k < m \le \frac{k}{\alpha^2}, \\ \frac{1}{2}cm\alpha, & \text{when } m \ge \frac{k}{\alpha^2} . \end{cases} \quad (7)$$

In order to prove the utility bounds, we also need the following (weak) version of the result of [38], which is sufficient to prove the sample complexity bound for constant error probability.

**Lemma 9.** *There is a constant $C > 0$, such that when $m > C\sqrt{k}/\alpha^2$, then for $X_1^m \sim p$, where either $p = u[k]$, or $d_{TV}(p, u[k]) \ge \alpha$,*

$$\Pr\left(|Z(X_1^m) - \mathbb{E}\left[Z(X_1^m)\right]| > \frac{2\mathbb{E}\left[Z(X_1^m)\right]}{3}\right) < 0.01.$$

The proof of this result is in Appendix C.1.

We now proceed to prove the sample complexity bounds. Assume that $m > C\sqrt{k}/\alpha^2$, so Lemma 9 holds. Suppose $\varepsilon$ be any real number such that $\varepsilon|\mathbb{E}\left[Z(X_1^m)\right]| > 3\log 100$. Let $\mathcal{A}(X_1^m)$ be the output of Algorithm 1. Denote the output by 1 when $\mathcal{A}(X_1^m)$ is "$p \neq u[k]$", and 0 otherwise. Consider the case when $X_1^m \sim p$, and $d_{TV}(p, u[k]) \geq \alpha$. Then,

$$
\begin{aligned}
\Pr\left(\mathcal{A}(X_1^m) = 1\right) &\geq \Pr\left(\mathcal{A}(X_1^m) = 1 \text{ and } Z(X_1^m) > \frac{\mathbb{E}\left[Z(X_1^m)\right]}{3}\right) \\
&= \Pr\left(Z(X_1^m) > \frac{\mathbb{E}\left[Z(X_1^m)\right]}{3}\right) \cdot \Pr\left(\mathcal{A}(X_1^m) = 1 | Z(X_1^m) > \frac{\mathbb{E}\left[Z(X_1^m)\right]}{3}\right) \\
&\geq 0.99 \cdot \Pr\left(B\left(\sigma(\varepsilon \cdot \frac{\mathbb{E}\left[Z(X_1^m)\right]}{3})\right) = 1\right) \\
&\geq 0.99 \cdot 0.99 \geq 0.9,
\end{aligned}
$$

where the last step uses that $\varepsilon\mathbb{E}\left[Z(X_1^m)\right]/3 > \log 100$, along with Lemma 4. The case of $p = u[k]$ follows from the same argument.

Therefore, the algorithm is correct with probability at least 0.9, whenever, $m > C\sqrt{k}/\alpha^2$, and $\varepsilon|\mathbb{E}\left[Z(X_1^m)\right]| > 3\log 100$. By (7), note that $\varepsilon|\mathbb{E}\left[Z(X_1^m)\right]| > 3\log 100$ is satisfied when,

$$
\begin{aligned}
c\alpha^2 \cdot m^2/k &\geq (6\log 100)/\varepsilon, \quad \text{for } m \leq k, \\
c\alpha^2 \cdot m^{3/2}/k^{1/2} &\geq (6\log 100)/\varepsilon, \quad \text{for } k < m \leq k/\alpha^2, \\
c\alpha \cdot m &\geq (6\log 100)/\varepsilon, \quad \text{for } m \geq k/\alpha^2.
\end{aligned}
$$

This gives the upper bounds for all the three regimes of $m$.

## C.1 Proof of Lemma 9

In order to prove the lemma, we need the following lemma, which is proved in [38].

**Lemma 10.** *(Bernstein version of McDiarmid's inequality) Let $Y_1^m$ be independent random variables taking values in the set $\mathcal{Y}$. Let $f : \mathcal{Y}^m \to \mathbb{R}$ be a function of $Y_1^m$ so that for every $j \in [m]$, and $y_1, ...y_m, y_j' \in \mathcal{Y}$, we have that:*

$$
\left|f(y_1, ...y_j, ...y_m) - f(y_1, ..., y_j', ...y_m)\right| \leq B,
$$

*Then we have*

$$
\Pr\left(f - \mathbb{E}\left[f\right] \geq z\right) \leq \exp\left(\frac{-2z^2}{mB^2}\right).
$$

*In addition, if for each $j \in [m]$ and $y_1, ...y_{j-1}, y_{j+1}, ...y_m$ we have that*

$$
\mathrm{Var}_{Y_j}[f(y_1, ...y_j, ...y_m)] \leq \sigma_j^2,
$$

*then we have*

$$
\Pr\left(f - \mathbb{E}\left[f\right] \geq z\right) \leq \exp\left(\frac{-z^2}{\sum_{j=1}^m \sigma_j^2 + 2Bz/3}\right).
$$

The statistic we use $Z(X_1^m)$ has sensitivity at most 1, hence we can use $B = 1$ in Lemma 10.

We first consider the case when $k < m \leq \frac{k}{\alpha^2}$. When $p = u[k]$, we get $\mathbb{E}\left[Z(X_1^m)\right] = -\frac{1}{2}cm\alpha^2 \cdot \sqrt{\frac{m}{k}}$, then by the first part of Lemma 10,

$$
\begin{aligned}
\Pr\left(Z(X_1^m) > \frac{\mathbb{E}\left[Z(X_1^m)\right]}{3}\right) &= \Pr\left(Z(X_1^m) > -\frac{1}{6}cm\alpha^2 \cdot \sqrt{\frac{m}{k}}\right) \\
&\leq \Pr\left(Z(X_1^m) - \mathbb{E}\left[Z(X_1^m)\right] > \frac{2}{3}cm\alpha^2 \cdot \sqrt{\frac{m}{k}}\right) \\
&\leq \exp\left(-\frac{8c^2 m^2 \alpha^4}{9k}\right). \quad\quad\quad (8)
\end{aligned}
$$

Therefore, there is a $C_1$ such that if $m \geq C_1 \sqrt{k}/\alpha^2$, then under the uniform distribution $\Pr\left(Z(X_1^m) > \frac{\mathbb{E}[Z(X_1^m)]}{3}\right)$ is at most $1/100$. The non-uniform distribution part is similar and we omit the case.

Then we consider the case when $\frac{k}{\alpha^2} < m$. When $p = u[k]$, we get $\mathbb{E}[Z(X_1^m)] = -\frac{1}{2}cm\alpha$, then also by the first part of Lemma 10,

$$\Pr\left(Z(X_1^m) > \frac{\mathbb{E}[Z(X_1^m)]}{3}\right) = \Pr\left(Z(X_1^m) > -\frac{1}{6}cm\alpha\right)$$
$$\leq \Pr\left(Z(X_1^m) - \mathbb{E}[Z(X_1^m)] > \frac{2}{3}cm\alpha\right)$$
$$\leq \exp\left(-\frac{8c^2m\alpha^2}{9}\right).$$

Using the same argument we can show that there is a constant $C_2$ such that for $m \geq C_2/\alpha^2$, then under the uniform distribution $\Pr\left(Z(X_1^m) > \frac{\mathbb{E}[Z(X_1^m)]}{3}\right)$ is at most $1/100$. The case of non-uniform distribution is omitted because of the same reason.

At last we consider the case when $m \leq k$. In this case we need another result proved in [38]:

$$\mathrm{Var}_{X_j}[Z(x_1, x_2, ..., X_j, .., x_m)] \leq \frac{m}{k}, \forall j, x_1, x_2, ..., x_{j-1}, x_{j+1}, ...x_n.$$

When $p = u[k]$, we get $\mathbb{E}[Z(X_1^m)] = -\frac{1}{2}ck\alpha^2 \cdot \frac{m^2}{k^2}$, then by the second part of Lemma 10,

$$\Pr\left(Z(X_1^m) > \frac{\mathbb{E}[Z(X_1^m)]}{3}\right) = \Pr\left(Z(X_1^m) > -\frac{1}{6}c\alpha^2 \cdot \frac{m^2}{k}\right)$$
$$\leq \Pr\left(Z(X_1^m) - \mathbb{E}[Z(X_1^m)] > \frac{2}{3}c\alpha^2 \cdot \frac{m^2}{k}\right)$$
$$\leq \exp\left(\frac{-\frac{4}{9}c^2\alpha^4\frac{m^4}{k^2}}{\frac{m^2}{k} + \frac{4}{9}c\alpha^2\frac{m^2}{k}}\right)$$
$$\leq \exp\left(-\frac{2}{9}c\alpha^4\frac{m^2}{k}\right).$$

Therefore, there is a $C_3$ such that if $m \geq C_3\sqrt{k}/\alpha^2$, then under the uniform distribution $\Pr\left(Z(X_1^m) > \frac{\mathbb{E}[Z(X_1^m)]}{3}\right)$ is at most $1/100$. The case of non-uniform distribution is similar and is omitted.

Therefore, if we take $C = \max\{C_1, C_2, C_3\}$, we prove the result in the lemma.

## D  Proof of Lemma 7

**D.1**  $m \leq k, \min\{\frac{m^2}{k}, \frac{m^{3/2}}{k^{1/2}}\} = \frac{m^2}{k}$,

Before proving the lemma, we consider an example that will provide insights and tools to analyze the distributions $Q_1$, and $Q_2$. Let $t \in \mathbb{N}$. Let $P_2$ be the following distribution over $\{0,1\}^t$:

- Select $b \in \{\frac{1}{2} - \alpha, \frac{1}{2} + \alpha\}$ with equal probability.
- Output $t$ independent samples from $B(b)$.

Let $P_1$ be the distribution over $\{0,1\}^t$ that outputs $t$ independent samples from $B(0.5)$.

When $t = 1$, $P_1$ and $P_2$ both become $B(0.5)$. For t=2, $P_1(00) = P_1(11) = \frac{1}{4} + \alpha^2$, and $P_1(10) = P_1(01) = \frac{1}{4} - \alpha^2$, and $d_{TV}(P_1, P_2)$ is $2\alpha^2$. A slightly general result is the following:

**Lemma 11.** *For $t = 1$, $d_{TV}(P_1, P_2) = 0$ and for $t \geq 2$, $d_{TV}(P_1, P_2) \leq 2t\alpha^2$.*

*Proof.* Consider any sequence $X_1^t$ that has $t_0$ zeros, and $t_1 = t - t_0$ ones. Then,

$$P_1(X_1^t) = \binom{t}{t_0} \frac{1}{2^t},$$

and

$$P_2(X_1^t) = \binom{t}{t_0} \frac{1}{2^t} \left( \frac{(1-2\alpha)^{t_0}(1+2\alpha)^{t_1} + (1+2\alpha)^{t_0}(1-2\alpha)^{t_1}}{2} \right).$$

The term in the parentheses above is minimized when $t_0 = t_1 = t/2$. In this case,

$$P_2(X_1^t) \geq P_1(X_1^t) \cdot (1+2\alpha)^{t/2}(1-2\alpha)^{t/2} = P_1(X_1^t) \cdot (1-4\alpha^2)^{t/2}.$$

Therefore,

$$d_{TV}(P_1, P_2) = \sum_{P_1 > P_2} P_1(X_1^t) - P_2(X_1^t) \leq \sum_{P_1 > P_2} P_1(X_1^t)\left( 1 - (1-4\alpha^2)^{t/2} \right) \leq 2t\alpha^2,$$

where we used the Weierstrass Product Inequality, which states that $1 - tx \leq (1-x)^t$ proving the total variation distance bound. $\qquad\square$

As a corollary this implies:

**Lemma 12.** *There is a coupling between $X_1^t$ generated from $P_1$ and $Y_1^t$ from $P_2$ such that* $\mathbb{E}\left[d_H(X_1^t, Y_1^t)\right] \leq t \cdot d_{TV}(P_1, P_2) \leq 4(t^2 - t)\alpha^2.$

*Proof.* Observe that $\sum_{X_1^t} \min\{P_1(X_1^t), P_2(X_1^t)\} = 1 - d_{TV}(P_1, P_2)$. Consider the following coupling between $P_1$, and $P_2$. Suppose $X_1^t$ is generated by $P_1$, and let $R$ be a $U[0,1]$ random variable.

1. $R < 1 - d_{TV}(P_1, P_2)$ Generate $X_1^t$ from the distribution that assigns probability $\frac{\min\{P_1(X_1^t), P_2(X_1^t)\}}{1 - d_{TV}(P_1, P_2)}$ to $X_1^t$. Output $(X_1^t, X_1^t)$.
2. $R \geq 1 - d_{TV}(P_1, P_2)$ Generate $X_1^t$ from the distribution that assigns probability $\frac{P_1(X_1^t) - \min\{P_1(X_1^t), P_2(X_1^t)\}}{d_{TV}(P_1, P_2)}$ to $X_1^t$, and $Y_1^t$ from the distribution that assigns probability $\frac{P_2(Y_1^t) - \min\{P_1(Y_1^t), P_2(Y_1^t)\}}{d_{TV}(P_1, P_2)}$ to $Y_1^t$ independently. Then output $(X_1^t, Y_1^t)$.

To prove the coupling, note that the probability of observing $X_1^t$ is

$$(1 - d_{TV}(P_1, P_2)) \cdot \frac{\min\{P_1(X_1^t), P_2(X_1^t)\}}{1 - d_{TV}(P_1, P_2)} + d_{TV}(P_1, P_2) \cdot \frac{P_1(X_1^t) - \min\{P_1(X_1^t), P_2(X_1^t)\}}{d_{TV}(P_1, P_2)} = P_1(X_1^t).$$

A similar argument gives the probability of $Y_1^t$ to be $P_2(Y_1^t)$.

Then $\mathbb{E}\left[d_H(X_1^t, Y_1^t)\right] \leq t \cdot d_{TV}(P_1, P_2) = 2t^2\alpha^2 \leq 4(t^2 - t)\alpha^2$ when $t \geq 2$, and when $t = 1$, the distributions are identical and the Hamming distance of the coupling is equal to zero. $\quad\square$

We now have the tools to prove Lemma 7 for $m \leq k$.

*Proof of Lemma 7 for $m \leq k$.* The following is a coupling between $Q_1$ and $Q_2$:

1. Generate $m$ samples $Z_1^m$ from a uniform distribution over $[k/2]$.
2. For $j \in [k/2]$, let $T_j \subseteq [m]$ be the set of locations where $j$ appears. Note that $|T_j| = M_j(Z_1^m)$.
3. To generate samples from $Q_1$:
   - Generate $|T_j|$ samples from a uniform distribution over $\{2j-1, 2j\}$, and replace the symbols in $T_j$ with these symbols.
4. To generate samples from $Q_2$:
   - Similar to the construction of $P_1$ earlier in this section, consider two distributions over $\{2j-1, 2j\}$ with bias $\frac{1}{2} - \alpha$, and $\frac{1}{2} + \alpha$.
   - Pick one of these distributions at random.
   - Generate $|T_j|$ samples from it over $\{2j-1, 2j\}$, and replace the symbols in $T_j$ with these symbols.

From this process the coupling between $Q_1$, and $Q_2$ is also clear:

- Given $X_1^m$ from $Q_2$, for each $j \in [k/2]$ find all locations $\ell$ such that $X_\ell = 2j - 1$, or $X_\ell = 2j$. Call this set $T_j$.
- Perform the coupling between $P_2$ and $P_1$ from Lemma 12, after replacing $\{0,1\}$ with $\{2j - 1, 2j\}$.

Using the coupling defined above, by the linearity of expectations, we get:

$$\mathbb{E}\left[d_H(X_1^m, Y_1^m)\right] = \sum_{j=1}^{k/2} \mathbb{E}\left[d_H(X_1^{|T_j|}, Y_1^{|T_j|})\right]$$

$$= \frac{k}{2} \mathbb{E}\left[d_H(X_1^R, Y_1^R)\right]$$

$$\leq \frac{k}{2} \cdot \mathbb{E}\left[4\alpha^2(R^2 - R)\right],$$

where $R$ is a binomial random variable with parameters $m$ and $2/k$. Now, a simple exercise computing Binomial moments shows that for $X \sim Bin(n,s)$, $\mathbb{E}\left[X^2 - X\right] = s^2(n^2 - n) \leq n^2s^2$. This implies that

$$\mathbb{E}\left[R^2 - R\right] \leq \frac{4m^2}{k^2}.$$

Plugging this, we obtain

$$\mathbb{E}\left[d_H(X_1^m, Y_1^m)\right] \leq \frac{k}{2} \cdot \frac{16\alpha^2 m^2}{k^2} = \frac{8m^2\alpha^2}{k},$$

proving the claim. $\qquad\square$

## D.2 $\quad k \leq m \leq k/\alpha^2, \min\{\frac{m^2}{k}, \frac{m^{3/2}}{k^{1/2}}\} = \frac{m^{3/2}}{k^{1/2}}$

Lemma 11 holds for all values of $t$, and $\alpha$. The lemma can be strengthened for cases where $\alpha$ is small.

**Lemma 13.** *Let $P_1$, and $P_2$ be the distributions over $\{0,1\}^t$ defined in the last section. There is a coupling between $X_1^t$ generated by $P_1$, and $Y_1^t$ by $P_2$ such that*

$$\mathbb{E}\left[d_H(X_1^t, Y_1^t)\right] \leq C \cdot (\alpha^2 t^{3/2} + \alpha^4 t^{5/2} + \alpha^5 t^3).$$

### D.2.1 Proof of Lemma 7 assuming Lemma 13

Given the coupling we defined in Appendix D.2.2 for proving Lemma 13, the coupling between $Q_1$, and $Q_2$ uses the same technique in the last section for $m \leq k$.

- Given $X_1^m$ from $Q_2$, for each $j \in [k/2]$ find all locations $\ell$ such that $X_\ell = 2j - 1$, or $X_\ell = 2j$. Call this set $T_j$.
- Perform the coupling in Appendix D.2.2 between $P_2$ and $P_1$ on $T_j$, after replacing $\{0,1\}$ with $\{2j - 1, 2j\}$.

Using the coupling defined above, by the linearity of expectations, we get:

$$\mathbb{E}\left[d_H(X_1^m, Y_1^m)\right] = \sum_{j=1}^{k/2} \mathbb{E}\left[d_H(X_1^{|T_j|}, Y_1^{|T_j|})\right]$$

$$= \frac{k}{2} \mathbb{E}\left[d_H(X_1^R, Y_1^R)\right]$$

$$\leq \frac{k}{2} \cdot \mathbb{E}\left[64 \cdot \left(\alpha^4 R^{5/2} + \alpha^2 R^{3/2} + \alpha^5 R^3\right)\right],$$

where $R \sim \text{Bin}(m, 2/k)$.

We now bound the moments of Binomial random variables. The bound is similar in flavor to [65, Lemma 3] for Poisson random variables.

**Lemma 14.** *Suppose $\frac{m}{k} > 1$, and $Y \sim \mathrm{Bin}(m, \frac{1}{k})$, then for $\gamma \geq 1$, there is a constant $C_\gamma$ such that*

$$\mathbb{E}\left[Y^\gamma\right] \leq C_\gamma \left(\frac{m}{k}\right)^\gamma.$$

*Proof.* For integer values of $\gamma$, this directly follows from the moment formula for Binomial distribution [66], and for other $\gamma \geq 1$, by Jensen's Inequality

$$\mathbb{E}\left[Y^\gamma\right] \leq \mathbb{E}\left[\left(Y^{\lceil\gamma\rceil}\right)^{\frac{\gamma}{\lceil\gamma\rceil}}\right] \leq \mathbb{E}\left[\left(Y^{\lceil\gamma\rceil}\right)\right]^{\frac{\gamma}{\lceil\gamma\rceil}} \leq \left(C_{\lceil\gamma\rceil}\mathbb{E}\left[Y\right]^{\lceil\gamma\rceil}\right)^{\frac{\gamma}{\lceil\gamma\rceil}} = C'(\mathbb{E}\left[Y\right])^\gamma,$$

proving the lemma. $\qquad\square$

Therefore, letting $C = \max\{C_{5/2}, C_3, C_{3/2}\}$, we obtain

$$\mathbb{E}\left[d_H(X_1^m, Y_1^m)\right] \leq 32kC \cdot \left(\alpha^4\left(\frac{m}{k}\right)^{5/2} + \alpha^2\left(\frac{m}{k}\right)^{3/2} + \alpha^5\left(\frac{m}{k}\right)^3\right).$$

Now, notice $\alpha\sqrt{\frac{m}{k}} < 1$. Plugging this,

$$\mathbb{E}\left[d_H(X_1^m, Y_1^m)\right] \leq 32C \cdot k \cdot \left(\alpha^4\left(\frac{m}{k}\right)^{5/2} + \alpha^2\left(\frac{m}{k}\right)^{3/2} + \alpha^5\left(\frac{m}{k}\right)^3\right)$$

$$= 32C \cdot k\alpha^2 \cdot \left(\alpha^2\frac{m}{k} \cdot \left(\frac{m}{k}\right)^{3/2} + \left(\frac{m}{k}\right)^{3/2} + \alpha^3\left(\frac{m}{k}\right)^{3/2}\left(\frac{m}{k}\right)^{3/2}\right)$$

$$\leq 96C \cdot k\left(\frac{m}{k}\right)^{3/2},$$

completing the argument.

### D.2.2 Proof of Lemma 13

To prove Lemma 13, we need a few lemmas first:

**Definition 7.** A random variable $Y_1$ is said to stochastically dominate $Y_2$ if for all $t$, $\Pr\left(Y_1 \geq t\right) \geq \Pr\left(Y_2 \geq t\right)$.

**Lemma 15.** *Suppose $N_1 \sim \mathrm{Bin}(t, \frac{1}{2}), N_2 \sim \frac{1}{2}\mathrm{Bin}(t, \frac{1+\alpha}{2}) + \frac{1}{2}\mathrm{Bin}(t, \frac{1-\alpha}{2})$. Then $Z_2 = \max\{N_2, t - N_2\}$ stochastically dominates $Z_1 = \max\{N_1, t - N_1\}$.*

*Proof.*

$$\Pr\left(Z_2 \geq l\right) = \sum_{i=0}^{t-l} \binom{t}{i}\left[\left(\frac{1+\alpha}{2}\right)^i\left(\frac{1-\alpha}{2}\right)^{t-i} + \left(\frac{1-\alpha}{2}\right)^i\left(\frac{1+\alpha}{2}\right)^{t-i}\right],$$

$$\Pr\left(Z_1 \geq l\right) = 2 \cdot \sum_{i=0}^{t-l} \binom{t}{i}\left(\frac{1}{2}\right)^t.$$

Define $F(l) = \Pr\left(Z_2 \geq l\right) - \Pr\left(Z_1 \geq l\right)$. What we need to show is $F(l) \geq 0, \forall l \geq \frac{t}{2}$. First we observe that $\Pr\left(Z_2 \geq \frac{t}{2}\right) = \Pr\left(Z_1 \geq \frac{t}{2}\right) = 1$ and $\Pr\left(Z_2 \geq t\right) = (\frac{1+\alpha}{2})^t + (\frac{1-\alpha}{2})^t \geq 2(\frac{1}{2})^t = \Pr\left(Z_1 \geq t\right)$. Hence $F(\frac{t}{2}) = 0, F(t) > 0$. Let

$$f(l) = F(l+1) - F(l) = -\binom{t}{l}\left[\left(\frac{1+\alpha}{2}\right)^l\left(\frac{1-\alpha}{2}\right)^{t-l} + \left(\frac{1-\alpha}{2}\right)^l\left(\frac{1+\alpha}{2}\right)^{t-l} - 2\left(\frac{1}{2}\right)^t\right].$$

Let $g(x) = \left(\frac{1+\alpha}{2}\right)^x\left(\frac{1-\alpha}{2}\right)^{t-x} + \left(\frac{1-\alpha}{2}\right)^x\left(\frac{1+\alpha}{2}\right)^{t-x} - 2\left(\frac{1}{2}\right)^t, x \in [t/2, t]$, then

$$\frac{dg(x)}{dx} = \ln\left(\frac{1+\alpha}{1-\alpha}\right) \cdot \left[\left(\frac{1+\alpha}{2}\right)^x\left(\frac{1-\alpha}{2}\right)^{t-x} - \left(\frac{1-\alpha}{2}\right)^x\left(\frac{1+\alpha}{2}\right)^{t-x}\right] \geq 0.$$

We know $g(t/2) < 0, g(t) > 0$, hence $\exists x^*, s.t. g(x) \leq 0, \forall x < x^*$ and $g(x) \geq 0, \forall x > x^*$. Because $f(l) = -\binom{t}{l}g(l)$, hence $\exists l^*, s.t. f(l) \leq 0, \forall l \geq l^*$ and $f(l) \geq 0, \forall l < l^*$. Therefore, $F(l)$ first increases and then decreases, which means $F(l)$ achieves its minimum at $\frac{t}{2}$ or $t$. Hence $F(l) \geq 0$, completing the proof. $\qquad\square$

For stochastic dominance, the following definition [67] will be useful.

**Definition 8.** A coupling $(X', Y')$ is a monotone coupling if $\Pr(X' \geq Y') = 1$.

The following lemma states a nice relationship between stochastic dominance and monotone coupling, which is provided as Theorem 7.9 in [67]

**Lemma 16.** *Random variable $X$ stochastically dominates $Y$ if and only if there is a monotone coupling between $(X', Y')$ with $\Pr(X' \geq Y') = 1$.*

By Lemma 16, there is a monotone coupling between $Z_1 = \max\{N_1, t - N_1\}$ and $Z_2 = \max\{N_2, t - N_2\}$. Suppose the coupling is $P^c_{Z_1, Z_2}$, we define the coupling between $X_1^t$ and $Y_1^t$ as following:

1. Generate $X_1^t$ according to $P_1$ and count the number of one's in $X_1^t$ as $n_1$.
2. Generate $n_2$ according to $P^c[Z_2|Z_1 = \max\{n_1, t - n_1\}]$.
3. If $n_1 > t - n_1$, choose $n_2 - n_1$ of the zero's in $X_1^t$ uniformly at random and change them to one's to get $Y_1^t$.
4. If $n_1 < t - n_1$, choose $n_2 - (t - n_1)$ of the one's in $X_1^t$ uniformly at random and change them to zero's to get $Y_1^t$.
5. If $n_1 = t - n_1$, break ties uniformly at random and do the corresponding action.
6. Output $(X_1^t, Y_1^t)$.

Since the coupling is monotone, and $d_H(X_1^t, Y_1^t) = Z_2 - Z_1$ for every pair of $(X_1^t, Y_1^t)$, we get:

$$\mathbb{E}\left[d_H(X_1^t, Y_1^t)\right] = \mathbb{E}\left[\max\{N_2, t - N_2\}\right] - \mathbb{E}\left[\max\{N_1, t - N_1\}\right].$$

Hence, to show lemma 13, it suffices to show the following lemma:

**Lemma 17.** *Suppose $N_1 \sim \text{Bin}(t, \frac{1}{2})$, $N_2 \sim \frac{1}{2}\text{Bin}(t, \frac{1+\alpha}{2}) + \frac{1}{2}\text{Bin}(t, \frac{1-\alpha}{2})$.*

$$\mathbb{E}\left[\max\{N_2, t - N_2\}\right] - \mathbb{E}\left[\max\{N_1, t - N_1\}\right] < C \cdot (\alpha^2 t^{3/2} + \alpha^4 t^{5/2} + \alpha^5 t^3)$$

*Proof.*

$$\mathbb{E}\left[\max\{N_2, t - N_2\}\right]$$

$$= \sum_{0 \leq \ell \leq t/2} (t/2 + \ell)\binom{t}{\frac{t}{2} - \ell}\left(\left(\frac{1-\alpha}{2}\right)^{\frac{t}{2}-\ell}\left(\frac{1+\alpha}{2}\right)^{\frac{t}{2}+\ell} + \left(\frac{1+\alpha}{2}\right)^{\frac{t}{2}-\ell}\left(\frac{1-\alpha}{2}\right)^{\frac{t}{2}+\ell}\right)$$

$$= \frac{t}{2} + \sum_{0 \leq \ell \leq t/2} \ell\binom{t}{\frac{t}{2} - \ell}\left(\left(\frac{1-\alpha}{2}\right)^{\frac{t}{2}-\ell}\left(\frac{1+\alpha}{2}\right)^{\frac{t}{2}+\ell} + \left(\frac{1+\alpha}{2}\right)^{\frac{t}{2}-\ell}\left(\frac{1-\alpha}{2}\right)^{\frac{t}{2}+\ell}\right).$$

Consider a fixed value of $t$. Let

$$f(\alpha) = \sum_{0 \leq \ell \leq t/2} \ell\binom{t}{\frac{t}{2} - \ell}\left(\left(\frac{1-\alpha}{2}\right)^{\frac{t}{2}-\ell}\left(\frac{1+\alpha}{2}\right)^{\frac{t}{2}+\ell} + \left(\frac{1+\alpha}{2}\right)^{\frac{t}{2}-\ell}\left(\frac{1-\alpha}{2}\right)^{\frac{t}{2}+\ell}\right).$$

The first claim is that this expression is minimized at $\alpha = 0$. This is because of the monotone coupling between $Z_1$ and $Z_2$, which makes $\mathbb{E}[Z_2] \geq \mathbb{E}[Z_1]$. This implies that $f'(0) = 0$, and by intermediate value theorem, there is $\beta \in [0, \alpha]$, such that

$$f(\alpha) = f(0) + \frac{1}{2}\alpha^2 \cdot f''(\beta). \tag{9}$$

We will now bound this second derivative. To further simplify, let

$$g(\alpha) = \left(\frac{1-\alpha}{2}\right)^{\frac{t}{2}-\ell}\left(\frac{1+\alpha}{2}\right)^{\frac{t}{2}+\ell} + \left(\frac{1+\alpha}{2}\right)^{\frac{t}{2}-\ell}\left(\frac{1-\alpha}{2}\right)^{\frac{t}{2}+\ell}.$$

Differentiating $g(\alpha)$, twice with respect to $\alpha$, we obtain,

$$g''(\alpha) = \frac{1}{16} \cdot \left(\alpha^2(t^2 - t) - 4\alpha\ell(t-1) + 4\ell^2 - t\right)\left(\frac{1-\alpha}{2}\right)^{\frac{t}{2}-\ell-2}\left(\frac{1+\alpha}{2}\right)^{\frac{t}{2}+\ell-2}$$

$$+ \frac{1}{16} \cdot \left(\alpha^2(t^2 - t) + 4\alpha\ell(t-1) + 4\ell^2 - t\right)\left(\frac{1+\alpha}{2}\right)^{\frac{t}{2}-\ell-2}\left(\frac{1-\alpha}{2}\right)^{\frac{t}{2}+\ell-2}.$$

Then $g''(\alpha)$ can be bound by,

$$g''(\alpha) \leq \frac{1}{16} \cdot (\alpha^2 t^2 + 4\ell^2) \left( \left( \frac{1-\alpha}{2} \right)^{\frac{t}{2}-\ell-2} \left( \frac{1+\alpha}{2} \right)^{\frac{t}{2}+\ell-2} + \left( \frac{1+\alpha}{2} \right)^{\frac{t}{2}-\ell-2} \left( \frac{1-\alpha}{2} \right)^{\frac{t}{2}+\ell-2} \right).$$

When $\alpha < \frac{1}{4}$, $(1-\alpha^2)^2 > \frac{1}{2}$, and we can further bound the above expression by

$$g''(\alpha) \leq 2 \cdot (\alpha^2 t^2 + 4\ell^2) \left( \left( \frac{1-\alpha}{2} \right)^{\frac{t}{2}-\ell} \left( \frac{1+\alpha}{2} \right)^{\frac{t}{2}+\ell} + \left( \frac{1+\alpha}{2} \right)^{\frac{t}{2}-\ell} \left( \frac{1-\alpha}{2} \right)^{\frac{t}{2}+\ell} \right).$$

Suppose $X$ is a $\text{Bin}(t, \frac{1+\beta}{2})$ distribution. Then, for any $\ell > 0$,

$$\Pr\left( \left| X - \frac{t}{2} \right| = \ell \right) = \binom{t}{\frac{t}{2}-\ell} \left( \left( \frac{1-\beta}{2} \right)^{\frac{t}{2}-\ell} \left( \frac{1+\beta}{2} \right)^{\frac{t}{2}+\ell} + \left( \frac{1+\beta}{2} \right)^{\frac{t}{2}-\ell} \left( \frac{1-\beta}{2} \right)^{\frac{t}{2}+\ell} \right).$$

Therefore, we can bound (9), by

$$f''(\beta) \leq 2 \cdot \left( \beta^2 t^2 \mathbb{E}\left[ \left| X - \frac{t}{2} \right| \right] + 4\mathbb{E}\left[ \left| X - \frac{t}{2} \right|^3 \right] \right).$$

For $X \sim \text{Bin}(m, r)$,

$$\mathbb{E}\left[ (X - mr)^2 \right] = mr(1-r) \leq \frac{m}{4}, \text{ and}$$

$$\mathbb{E}\left[ (X - mr)^4 \right] = mr(1-r)(3r(1-r)(m-2)+1) \leq 3\frac{m^2}{4}.$$

We bound each term using these moments,

$$\mathbb{E}\left[ \left| X - \frac{t}{2} \right| \right] \leq \mathbb{E}\left[ \left( X - \frac{t}{2} \right)^2 \right]^{1/2} = \left( t\frac{(1-\beta^2)}{4} + \left( \frac{t\beta}{2} \right)^2 \right)^{1/2} \leq \sqrt{t} + t\beta.$$

We similarly bound the next term,

$$\mathbb{E}\left[ \left| X - \frac{t}{2} \right|^3 \right] \leq \mathbb{E}\left[ \left( X - \frac{t}{2} \right)^4 \right]^{3/4}$$

$$\leq \mathbb{E}\left[ \left( X - \frac{t(1+\beta)}{2} + \frac{t\beta}{2} \right)^4 \right]^{3/4}$$

$$\leq 8 \left( \mathbb{E}\left[ \left( X - \frac{t(1+\beta)}{2} \right)^4 \right]^{3/4} + \left( \frac{t\beta}{2} \right)^3 \right)$$

$$\leq 8 \left( t^{3/2} + \left( \frac{t\beta}{2} \right)^3 \right),$$

where we use $(a+b)^4 \leq 8(a^4 + b^4)$.

Therefore,

$$f''(\beta) \leq 64 \cdot \left( \beta^2 t^{5/2} + t^{3/2} + (t\beta)^3 \right) \leq 64 \cdot \left( \alpha^2 t^{5/2} + t^{3/2} + (t\alpha)^3 \right).$$

As a consequence,

$$\mathbb{E}\left[ \max\{N_2, t - N_2\} \right] - \mathbb{E}\left[ \max\{N_1, t - N_1\} \right] = \alpha^2 f''(\beta) \leq 64 \cdot (\alpha^2 t^{3/2} + \alpha^4 t^{5/2} + \alpha^5 t^3).$$

completing the proof. □

# E  Proof of Theorem 4

## E.1  Closeness Testing – Upper Bounds

In this section, we will show that Algorithm 2 satisfies sample complexity upper bounds described in Theorem 4.

The results in [10] were proved under Poisson sampling, and we also use Poisson sampling, with only a constant factor effect on the number of samples for the same error probability. They showed the following bounds:

$$\mathbb{E}\left[Z(X_1^m, Y_1^m)\right] = 0 \text{ when } p = q, \tag{10}$$

$$\mathrm{Var}(Z(X_1^m, Y_1^m)) \leq 2\min\{k, m\} \text{ when } p = q, \tag{11}$$

$$\mathbb{E}\left[Z(X_1^m, Y_1^m)\right] \geq \frac{m^2\alpha^2}{4k + 2m} \text{ when } d_{TV}(p, q) \geq \alpha, \tag{12}$$

$$\mathrm{Var}(Z(X_1^m, Y_1^m)) \leq \frac{1}{1000}\mathbb{E}\left[Z(X_1^m, Y_1^m)\right]^2 \text{ when } p \neq q, \text{ and } m = \Omega\left(\frac{1}{\alpha^2}\right). \tag{13}$$

**Case 1:** $\alpha^2 > \frac{1}{\sqrt{k}}$, **and** $\alpha^2\varepsilon > \frac{1}{k}$. In this case, we will show that $S(\mathtt{CT}, k, \alpha, \varepsilon) = O\left(\frac{k^{2/3}}{\alpha^{4/3}} + \frac{k^{1/2}}{\alpha\sqrt{\varepsilon}}\right)$. In this case, $\frac{k^{2/3}}{\alpha^{4/3}} + \frac{k^{1/2}}{\alpha\sqrt{\varepsilon}} \leq 2k$.

We consider the case when $p = q$, then $\mathrm{Var}(Z(X_1^m, Y_1^m)) \leq 2\min\{k, m\}$. Let $\mathrm{Var}(Z(X_1^m, Y_1^m)) \leq cm$ for some constant $c$. By the Chebyshev's inequality,

$$
\begin{aligned}
\Pr\left(Z' > -\frac{1}{84} \cdot \frac{m^2\alpha^2}{4k + 2m}\right) &\leq \Pr\left(Z(X_1^m, Y_1^m) - \mathbb{E}\left[Z(X_1^m, Y_1^m)\right] > \frac{1}{3} \cdot \frac{m^2\alpha^2}{4k + 2m}\right) \\
&\leq \Pr\left(Z(X_1^m, Y_1^m) - \mathbb{E}\left[Z(X_1^m, Y_1^m)\right] > \frac{1}{3} \cdot \frac{m^2\alpha^2}{8k}\right) \\
&\leq \Pr\left(Z(X_1^m, Y_1^m) - \mathbb{E}\left[Z(X_1^m, Y_1^m)\right] > (cm)^{1/2} \cdot \frac{m^{3/2}\alpha^2}{24c^{1/2}k}\right) \\
&\leq 576c \cdot \frac{k^2}{m^3\alpha^4},
\end{aligned}
$$

where we used that $4k + 2m \leq 8k$.

Therefore, there is a $C_1$ such that if $m \geq C_1 k^{2/3}/\alpha^{4/3}$, then under $p = q$, $\Pr\left(Z' > -\frac{1}{84} \cdot \frac{m^2\alpha^2}{4k+2m}\right)$ is at most $1/100$. Now furthermore, if $\varepsilon \cdot m^2\alpha^2/(672k) > \log(20)$, then for all $Z' < -\frac{1}{84} \cdot \frac{m^2\alpha^2}{4k+2m}$, with probability at least $0.95$, the algorithm outputs the $p = q$. Combining the conditions, we obtain that there is a constant $C_2$ such that for $m = C_2\left(\frac{k^{2/3}}{\alpha^{4/3}} + \frac{k^{1/2}}{\alpha\sqrt{\varepsilon}}\right)$, with probability at least $0.9$, the algorithm outputs the correct answer when the input distributions satisfy $p = q$. The case of $d_{TV}(p, q) > \alpha$ distribution is similar and is omitted.

**Case 2:** $\alpha^2 < \frac{1}{\sqrt{k}}$, **or** $\alpha^2\varepsilon < \frac{1}{k}$. In this case, we will prove a bound of $O\left(\frac{\sqrt{k}}{\alpha^2} + \frac{1}{\alpha^2\varepsilon}\right)$ on the sample complexity. We still consider the case when $p = q$. We first note that when $\alpha^2 < \frac{1}{\sqrt{k}}$, or $\alpha^2\varepsilon < \frac{1}{k}$, then either $\frac{\sqrt{k}}{\alpha^2} + \frac{1}{\alpha^2\varepsilon} > k$. Hence we can assume that the sample complexity bound we aim for is at least $\Omega(k)$. So $\mathrm{Var}(Z(X_1^m, Y_1^m)) \leq ck$ for constant $c$. By the Chebyshev's inequality,

$$\Pr\left(Z' > -\frac{1}{84} \cdot \frac{m^2\alpha^2}{4k+2m}\right) \leq \Pr\left(Z(X_1^m, Y_1^m) - \mathbb{E}\left[Z(X_1^m, Y_1^m)\right] > \frac{1}{3} \cdot \frac{m^2\alpha^2}{4k+2m}\right)$$

$$\leq \Pr\left(Z(X_1^m, Y_1^m) - \mathbb{E}\left[Z(X_1^m, Y_1^m)\right] > \frac{1}{3} \cdot \frac{m\alpha^2}{6}\right)$$

$$\leq \Pr\left(Z(X_1^m, Y_1^m) - \mathbb{E}\left[Z(X_1^m, Y_1^m)\right] > (ck)^{1/2} \cdot \frac{m\alpha^2}{18c^{1/2}k^{1/2}}\right)$$

$$\leq 144 \cdot c \cdot \frac{k}{m^2\alpha^4}.$$

Therefore, there is a $C_1$ such that if $m \geq C_1 k^{1/2}/\alpha^2$, then under $p = q$, $\Pr\left(Z' > -\frac{1}{84} \cdot \frac{m^2\alpha^2}{4k+2m}\right)$ is at most $1/100$. In this situation, if $\varepsilon \cdot m\alpha^2/504 > \log(20)$, then for all $Z' < -\frac{1}{84} \cdot \frac{m^2\alpha^2}{4k+2m}$, with probability at least $0.95$, the algorithm outputs the $p = q$. Combining with the previous conditions, we obtain that there also exists a constant $C_2$ such that for $m = C_2\left(\frac{\sqrt{k}}{\alpha^2} + \frac{1}{\alpha^2\varepsilon}\right)$, with probability at least $0.9$, the algorithm outputs the correct answer when the input distribution is $p = q$. The case of $d_{TV}(p, q) > \alpha$ distribution is similar and is omitted.

### E.2 Closeness Testing – Lower Bounds

To show the lower bound part of Theorem 4, we need the following simple result.

**Lemma 18.** $S(\texttt{IT}, k, \alpha, \varepsilon) \leq S(\texttt{CT}, k, \alpha, \varepsilon)$.

*Proof.* Suppose we want to test identity with respect to $q$. Given $X_1^m$ from $p$, generate $Y_1^m$ independent samples from $q$. If $p = q$, then the two samples are generated by the same distribution, and otherwise they are generated by distributions that are at least $\varepsilon$ far in total variation. Therefore, we can simply return the output of an $(k, \alpha, \varepsilon)$-closeness testing algorithm on $X_1^m$, and $Y_1^m$. $\qquad\square$

By Lemma 18 we know that a lower bound for identity testing is also a lower bound on closeness testing.

We first consider the sparse case, when $\alpha^2 > \frac{1}{\sqrt{k}}$, and $\alpha^2\varepsilon > \frac{1}{k}$. In this case, we show that

$$S(\texttt{CT}, k, \alpha, \varepsilon) = \Omega\left(\frac{k^{2/3}}{\alpha^{4/3}} + \frac{\sqrt{k}}{\alpha\sqrt{\varepsilon}}\right).$$

When $\alpha > \frac{1}{k^{1/4}}$, $\frac{k^{2/3}}{\alpha^{4/3}}$ is the dominating term in the sample complexity $S(\texttt{CT}, k, \alpha) = \Theta\left(\frac{k^{2/3}}{\alpha^{4/3}} + \frac{\sqrt{k}}{\alpha^2}\right)$, giving us the first term. By Lemma 18 we know that a lower bound for identity testing is also a lower bound on closeness testing giving the second term, and the lower bound of Theorem 2 contains the second term as a summand.

In the dense case, when $\alpha^2 < \frac{1}{\sqrt{k}}$, or $\alpha^2\varepsilon < \frac{1}{k}$, we show that

$$S(\texttt{CT}, k, \alpha, \varepsilon, \delta) = \Omega\left(\frac{\sqrt{k}}{\alpha^2} + \frac{\sqrt{k}}{\alpha\sqrt{\varepsilon}} + \frac{1}{\alpha\varepsilon}\right).$$

In the dense case, using the non-private lower bounds of $\Omega\left(\frac{k^{2/3}}{\alpha^{4/3}} + \frac{\sqrt{k}}{\alpha^2}\right)$ along with the identity testing bound of sample complexity lower bounds of note that $\frac{\sqrt{k}}{\alpha\sqrt{\varepsilon}} + \frac{1}{\alpha\varepsilon}$ gives a lower bound of $\Omega\left(\frac{k^{2/3}}{\alpha^{4/3}} + \frac{\sqrt{k}}{\alpha^2} + \frac{\sqrt{k}}{\alpha\sqrt{\varepsilon}} + \frac{1}{\alpha\varepsilon}\right)$. However, in the dense case, it is easy to see that $\frac{k^{2/3}}{\alpha^{4/3}} = O\left(\frac{\sqrt{k}}{\alpha^2} + \frac{\sqrt{k}}{\alpha\sqrt{\varepsilon}}\right)$ giving us the bound.