[Reviews · NeurIPS 2018]

Reviewer 1



This paper studies the finite sample complexity of identity testing and closeness testing under the constraints of differential privacy. The paper introduces a technique based on probabilistic coupling useful to prove lower bounds on the sample complexity of some statistical tasks. It uses this technique to show lower bounds on uniformity testing which implies a lower bound on identity testing, and on closeness testing. The paper also provides upper bounds. Pros: +the general technique that the paper proposes, using probabilistic coupling to prove lower bounds, is neat and interesting and it could be applied to many other statistical tasks beyond the scope of this paper, +the paper finite sample complexity results clarify the picture both respect to the non-private setting, and with respect to previous works in the private setting, +the result showing that the reduction from identity testing to uniform testing in the non-private setting extends to the private setting is an interesting observation, +the paper is well written and gives enough details to understand the main results. Cons: -this is not the first paper using probabilistic coupling arguments in the setting of differential privacy and few related works are missing: [1] gives a lower bound proof for confidence intervals using a coupling argument, [2,3] characterize differential privacy through a coupling argument and use this characterization in program verification, [4] uses an implicit coupling argument in one of their main results. [1]Vishesh Karwa, Salil P. Vadhan: Finite Sample Differentially Private Confidence Intervals. ITCS 2018 [2]Gilles Barthe, Noémie Fong, Marco Gaboardi, Benjamin Grégoire, Justin Hsu, Pierre-Yves Strub: Advanced Probabilistic Couplings for Differential Privacy. CCS 2016 [3]Gilles Barthe, Marco Gaboardi, Benjamin Grégoire, Justin Hsu, Pierre-Yves Strub: Proving Differential Privacy via Probabilistic Couplings. LICS 2016 [4]Cynthia Dwork, Moni Naor, Omer Reingold, Guy N. Rothblum: Pure Differential Privacy for Rectangle Queries via Private Partitions. ASIACRYPT (2) 2015 #After author rfeedback Thanks for your answer. I agree with Reviewer #2, it would be great to have a description of the tests in the main part of the paper. I updated the list of references above on coupling.

Reviewer 2



This paper presents upper and lower bounds on the sample complexities of private versions of the identity and closeness testing problems. The upper and lower bounds for identity testing match for all parameter ranges. However, for the closeness testing problem, there's a gap of between the upper and lower bounds when the number of samples is larger than "k" (size of the support of the distributions). The results of this paper are interesting and fundamental. The authors privatize recent test statistics: DGPP17 (for identity testing) and CDVV14 (for closeness testing) to get their upper bounds. The lower bound for identity testing is based on Le Cam's method and the same lower bound is used for closeness testing. The paper is well written and organized. Comments: 1. In lines 129 and 130: "Both [CDK17], and [ADR17] consider only pure-differential privacy, which are a special case of our results." But isn't that okay since you argue that pure-DP is enough? 2. It would be nice if the test statistics (and their private versions) are discussed in the main writeup. Perhaps you can move the proof of Theorem 1 (which seems very basic) to the appendix to make some space? 3. It would be nice if you can end with a concluding section that not only recaps the main results but also discusses some interesting extensions and future research directions. Overall, I think this is a good paper and recommend it for publication. #After author rfeedback Thanks for your detailed response. I am glad to see that you have addressed my points. Looking forward to reading your final version :)

Reviewer 3



This paper studies two basic testing problems for distributions over finite domains. The first is the identity testing problem, where given a known distribution q and i.i.d. samples from p, the goal is to distinguish between the case where p = q and the case where the distributions are far in total variation distance. A second, closely related problem, is the closeness testing problem where q is also taken to be an unknown distribution accessed via i.i.d. samples. This submission studies the sample complexity of solving these problems subject to differential privacy. For identity testing, it gives sample complexity upper and lower bounds which match up to constant factors. For closeness testing, it gives bounds which match in the "sparse" regime where the data domain is much larger than the accuracy/privacy parameters, and which match up to polynomial factors otherwise. The private identity testing problem that the authors consider had been studied in prior work of Cai, Diakonikolas, and Kamath, and this submission gives an improved (and simplified) upper bound. Private closeness testing appears not to have been studied before. (Independent work of Aliakbarpour, Diakonikolas, and Rubinfeld gives similar upper bounds which are complemented by experimental results, but does not study lower bounds.) The upper bounds are obtained by a) privatizing a reduction from identity testing to uniformity testing, due to Goldreich, and b) modifying recent uniformity and closeness testers to guarantee differential privacy. These modifications turn out to be relatively straightforward, as the test statistics introduced in these prior works have low sensitivity. Hence, private testers can be obtained by making randomized decisions based on these test statistics, instead of deterministically thresholding them. Lower bounds follow from a new coupling technique. The idea is to construct two distributions on finite samples that an accurate tester must be able to distinguish. However, if there is a coupling between these distributions where the samples have low expected Hamming distance, then differentially private algorithms will have a hard time telling them apart. This is a nice trick which is powerful enough to prove tight lower bounds, and extends generically to give lower bounds for (eps, delta)-DP. To summarize, the results are fairly interesting, deftly leverage recent advances in distribution testing, and introduce some nice new techniques. I think this is a solid paper for NIPS. After reading author feedback: I still believe this is a strong paper and should be accepted.